# Balanced Translocations Involving the *DMD* Gene as a Cause of Muscular Dystrophy in Female Children: A Description of Three Cases

**DOI:** 10.3390/ijms26199389

**Published:** 2025-09-25

**Authors:** Ekaterina O. Vorontsova, Aysylu Murtazina, Elena Zinina, Alexander V. Polyakov, Maria Sumina, Olga A. Rybakova, Dmitry Vlodavets, Dmitry Kazakov, Yulia Suvorova, Inna V. Sharkova, Nina A. Demina, Svetlana A. Repina, Vera A. Bulanova, Maria Antonova, Elena Dadali, Andrey V. Marakhonov, Nadezhda V. Shilova, Sergey I. Kutsev, Olga A. Shchagina

**Affiliations:** 1Research Centre for Medical Genetics, 115522 Moscow, Russia; 2State Healthcare Institution of Sverdlovsk Region, Clinical and Diagnostic Center «Mother’s and Child Health Protection», 620067 Yekaterinburg, Russia; 3Research Institute of Pediatrics and Pediatric Surgery Named After Academician Yu.E. Veltishchev Federal State Autonomous Educational Institution of the Russian National Research Medical University Named After N.I. Pirogov of the Russian Ministry of Health, 125412 Moscow, Russia; 4LLC “Biotech Campus”, 117437 Moscow, Russia; 5The Department of Nervous Diseases with Medical Genetics and Neurosurgery, Yaroslavl State Medical University, 150010 Yaroslavl, Russia

**Keywords:** DMD, girls, female, translocation, Duchenne muscular dystrophy, muscle MRI

## Abstract

Duchenne muscular dystrophy (DMD) is typically described in boys with a pathogenic variant in the *DMD*. However, in certain cases, females may also exhibit symptoms of this X-linked disorder. In the present study, the cause of Duchenne muscular dystrophy in three girls was reciprocal translocations t(X;2), t(X;12), and t(X;16), with breakpoints located within the *DMD* gene sequence. All patients had global development delay, predominantly proximal muscle weakness, calf muscle hypertrophy, and elevated creatine kinase levels up to 100 times the normal range (16,000–26,694 U/L). All underwent cardiac ultrasound and electromyography, and two of the girls also had muscle MRI data. After receiving negative results of MLPA aimed at the detection of *DMD* deletions and duplications, as well as the limb-girdle muscular dystrophy gene panel sequencing, the patients were referred to whole genome sequencing, which allowed to detect a translocation involving the short arm of the X chromosome and with breakpoints in the *DMD*. Karyotyping confirmed reciprocal translocations in all patients, with de novo status established in all three cases. The results of this study contribute to the understanding of clinical polymorphism and genetic heterogeneity of the disease, highlighting the importance of a comprehensive approach to genetic diagnostics in atypical cases.

## 1. Introduction

Muscular dystrophies caused by alterations in the *DMD* gene comprise a spectrum of X-linked muscle disorders: Duchenne muscular dystrophy (DMD), Becker muscular dystrophy, and X-linked dilated cardiomyopathy [1]. Although these conditions are described as X-linked recessive, symptoms may also manifest in female individuals. Female carriers of pathogenic variants in the *DMD* gene may show symptoms of the disorder, including muscle weakness, gait disturbances, cardiac problems, and elevated levels of creatine kinase (CK) and transaminases [2,3,4].

The spectrum of pathogenic *DMD* gene variants identified in symptomatic females does not significantly differ from that found in male patients and includes deletions in 68.5% of cases, duplications in 11%, and point variants in 20.6% [5]. In addition to deletions, duplications, and point variants, translocations involving the X chromosome with breakpoints within the *DMD* gene have also been described in females as a cause of muscular dystrophy [2,3,4,6,7]. The phenotype in such patients resembles the classical DMD phenotype observed in boys, manifesting in early childhood [2,3,4].

The current study provides a detailed description of the clinical features of muscular dystrophy in three girls with reciprocal translocations t(X;2), t(X;12), and t(X;16), with breakpoints located within the *DMD* gene sequence.

## 2. Results

### 2.1. Clinical Characteristics

The clinical picture in all three patients was similar. In two patients, the first signs of muscular dystrophy appeared before the age of three years; in one patient, clinical manifestations were detected at the age of five years. A detailed description of the clinical presentation, along with laboratory and instrumental findings, was provided in Table 1 and Table 2.

Patient N1 was a 4-year-old girl who had delayed speech development during the first year of life. She began speaking in short phrases after the age of 4. Divergent strabismus on the right side was noted from the age of 1. Behavioral peculiarities included limited social interaction; although she understood spoken language, she did not follow commands. Examination at the age of 4 revealed speech limited to single words and rare phrases, muscle weakness, easy fatigability, asymmetric calf hypertrophy (more prominent on the right) (Figure 1), and a positive Gowers’ sign. The follow-up examination at the age of 7 showed persistent symptoms and additional findings, including thigh muscle hypertrophy, pain in the thighs during walking, and delayed speech development. The six-minute walk test (6MWT) was first conducted at age 5 and measured 400 m; a repeat test at age 6 measured 375 m. NSAA at age 6 was 28 points.

Leg muscles EMG revealed a myogenic pattern of impairment. ECG showed incomplete right bundle block and decreased myocardial repolarization, manifested as flattened T waves in lead III. Cardiac US revealed a left ventricular ejection fraction (LVEF) of 61.4% (normal >60%), and the left ventricular end-diastolic diameter was at the upper limit of normal (LVEDD 37.4 mm; normal up to 37 mm). Brain and muscle MRI were performed at the age of 4 years (Figure 2). Brain MRI showed no abnormalities. While lower limb muscle T1-weighted sequences appeared normal, T2 DIXON sequences showed hyperintensity in the soleus and medial gastrocnemius muscles, suggestive of edema.

Patient N2 was a 6-year-old girl. At the age of 6 months, elevated liver enzyme (ALT, AST, LDH) levels and CK levels up to 11,000 U/L were noted. The patient began sitting at 7 months and started walking at 1 year and 2 months. At the age of 1 year, intermittent convergent strabismus on the left side was observed. Since the age of 2 years, speech development delay was noted, with difficulties in phrase speech; the patient attended a speech therapist and neuropsychologist. From the age of 3 years, the patient exhibited fatigue during physical activity and used Gowers’ sign when rising from the floor. Behavioral peculiarities were present: the child made limited social contact, understood spoken language, but did not follow commands. Clinical examination at the age of 6 years revealed poor vocabulary with speech consisting of single words, rarely short phrases. Muscle weakness, rapid fatigability, asymmetrical hypertrophy of the calf (D > S) and thigh muscles, Gowers’ maneuver, and asymmetrical winged scapula (D > S) were observed. EMG showed a myogenic pattern in lower limb muscles. ECG and cardiac ultrasound were performed. The NSAA score at the age of 6 years was 29 points.

Patient N3 was a 6-year-old girl with normal early motor development, she began walking at the age of 1 year. Since the age of 5 years, the parents began noticing changes in gait, particularly when climbing stairs, and calf muscle densification. Because of reduced exercise tolerance and difficulty ascending stairs, the girl was transferred to home-based education. Decreased academic performance was noted. Clinical examination at the age of 9 years revealed winged scapulae, gluteal and thigh muscles weakness, lumbar hyperlordosis, waddling gait, Gowers’ sign, and calf muscle hypertrophy. Needle EMG showed myogenic pattern in leg muscles. Cardiac US revealed no abnormalities. Lower limb muscle MRI, performed at the age of 7 and 8 years, showed a classic “trefoil with a single fruit” pattern, characterized by severe fatty replacement of the adductor magnus and biceps femoris, alongside notable sparing of the gracilis, sartorius, adductor longus, and semitendinosus muscles. Comparative analysis showed disease progression over one year, marked by increased fatty infiltration of the quadriceps muscles. All gluteal muscles exhibited severe fatty infiltration.

Within the lower legs, the peroneus longus and medial gastrocnemius muscles demonstrated less severe involvement (Figure 3).

### 2.2. Molecular Genetic Diagnostics

All patients were screened for point variants using a limb-girdle muscular dystrophy gene panel and for *DMD* gene rearrangements using MLPA.

No pathogenic variants were identified at this stage. Subsequently, WGS was performed in a “trio” format (biological parents and child) for two female patients, and in a “mono” format for one patient.

Analysis of WGS data in the three patients did not reveal any copy number variations or point variants (heterozygous for autosomal dominant inheritance; homozygous or compound heterozygous for autosomal recessive inheritance) in autosomal muscular dystrophy genes that could explain the clinical manifestations of muscular dystrophy in the girls. Structural variant analysis suggested potential translocations involving the X chromosome and one of the autosomes.

In patient N1, a reciprocal translocation of short arm of chromosome X and long arm of chromosome 2 was identified. The breakpoint on the X chromosome is located in intron 1 of the *DMD* gene (NC_000023.11:g.pter_31972130delins[NC_000002.12:g.pter_107014253inv]), with a single nucleotide duplication chrX:31972131 in the breakpoint region. On chromosome 2, a breakpoint region NC_000002.12:g.107014254_qterdelins[TTTTTTT;NC_000023.11:g.pter_31972131inv] is accompanied by a single nucleotide duplication chr2:107014253 and non-template T_7_-tract insertion no coding sequences were present in this region (Appendix A).

In addition to the structural rearrangement, WGS of patient N1 revealed a variant of uncertain clinical significance in exon 9 of the *COL6A3* gene (chr2:237371914G>A, NM_004369.4(*COL6A3*):c.4103C>T (p.Thr1368Met)) in a heterozygous state. This *COL6A3* variant was also found in the healthy father of patient N1.

In patient N2, a possible reciprocal translocation of short arm of chromosome X and long arm of chromosome 12 was identified. The breakpoint on the X chromosome is located in intron 44 of the *DMD* gene (NC_000023.11:g.pter_32042465delins[NC_000012.12:g.(q11-12)_qterinv]). It was not possible to determine the exact breakpoint on chromosome 12 due to the mapping of rearrangement-associated reads to an alpha-satellite repeat (NC_000012.12:g.(q11-q12)_qterdelins[NC_000023.11:g.32042468_qterinv]), which is typically localized in pericentromeric regions; no coding sequences were present in this region. The rearrangement also caused a dinucleotide deletion: NC_000023.11:g.32042466_32042467del. No other variants explaining the clinical findings were identified through whole genome sequencing.

As a result of “trio” whole genome sequencing in patient N3, a possible reciprocal translocation of short arm of chromosome X and either chromosomes 16 or 7 was detected. The breakpoint on the X chromosome is located within intron 16 of the *DMD* gene (NC_000023.11:g.pter_32561731delins[?]). The second breakpoint could not be precisely determined due to the mapping of rearrangement-associated reads to pericentromeric regions of several chromosomes, primarily chromosomes 16 and 7. The rearrangement also caused a single-nucleotide deletion: NC_000023.11:g.32561732del at the breakpoint site. No clinically significant genes were found in the mapped pericentromeric regions of chromosomes 16 and 7.

In addition to the identified chromosomal rearrangement, the patient was found to carry a previously reported pathogenic variant in the splice region of intron 12 of the *TRAPPC11* gene (chr4:183684059G>A) (NM_021942.6:c.1287+5G>A) in a heterozygous state [8]. This variant was inherited from the mother of proband 3. No second pathogenic or likely pathogenic variant was found, and the variant was classified as carriership of a single pathogenic variant in a gene associated with recessive disorders.

X-chromosome inactivation analysis was conducted on peripheral blood lymphocytes. All three patients had skewed X-chromosome inactivation. Patient N1 demonstrated skewed X-inactivation at a ratio of 96%:4%; Patient N2—98%:2%; Patient N3—90%:10%.

### 2.3. Cytogenetic Analysis

Reciprocal translocations were confirmed using the reference method. The results of standard cytogenetic analysis of the patients, as well as diagrams of the structural rearrangements, are presented in Figure 4.

For all three patients, karyotype analysis revealed that the translocations were balanced. It was determined that the karyotypes of the parents of patients N1, N2 and N3 were normal.

Thus, the karyotype of patient N1 is 46,XX,t(X;2)(p21.1;q12.3)dn, indicating a reciprocal translocation between the short arm of the X chromosome and the long arm of chromosome 2, arising de novo. The karyotype of patient N2 is 46,XX,t(X;12)(p21.1;q12)dn, indicating a reciprocal translocation between the short arm of the X chromosome and the long arm of chromosome 12, arising de novo. The karyotype of patient N3 is 46,X,t(X;16)(p21.1;p11.2)dn, indicating a reciprocal translocation between the short arm of the X chromosome and the short arm of chromosome 16, also arising de novo. In all three cases, the structural rearrangement involved the short arm of the X chromosome with a breakpoint at p21.1.

## 3. Discussion

Here we described three girls with muscular dystrophy caused by translocations with breakpoints located within the *DMD* gene.

To establish the cause of muscular dystrophy, initially, a panel of 31 limb-girdle muscular dystrophy genes was analyzed; no potentially causative variants were identified. WGS revealed point variants in the *COL6A3* and *TRAPPC11* genes in two patients.

In patient N1, a heterozygous variant was identified in the *COL6A3* gene. (NM_004369.4(*COL6A3*):c.4103C>T(p.Thr1368Met)), which is associated with autosomal dominant or autosomal recessive forms of Ullrich-Bethlem myopathy (OMIM 620726, OMIM 620728) or with autosomal recessive dystonia type 27 (OMIM 616411). However, the pattern of muscle involvement observed on MRI in patient N1 does not correspond to the specific pattern seen in collagenopathies, which are more commonly associated with an “outside-in” involvement of quadriceps femoris and calf muscles [9,10]. This variant was inherited from the patient’s healthy father.

A heterozygous variant in the *TRAPPC11* gene (NM_021942.6: c.1287+5G>A) was identified in patient N3. This gene is associated with the autosomal recessive limb-girdle muscular dystrophy type 18 (OMIM 615356). The pathogenesis of LGMD type 18 is based on a glycosylation defect, which manifests as either a muscular phenotype (resembling muscular dystrophy symptoms) or a neurovisceral phenotype. No second pathogenic variant in the *TRAPPC11* gene was found on WGS. The clinical picture of the proband does not correspond to the symptoms typically seen in congenital disorders of glycosylation [8,11].

The absence of second pathogenic variants in the *TRAPPC11* and *COL6A3* genes, the inheritance of the variants from healthy parents, and the incomplete clinical correspondence with the phenotypes described for muscular dystrophies caused by pathogenic variants in these genes allowed the identified heterozygous pathogenic variants to be interpreted as carrier status for autosomal recessive disorders in the patients.

In all three our cases, WGS revealed balanced translocations with breakpoints located within the *DMD* on the X chromosome. Therefore, the likely cause of the severe muscular dystrophy symptoms in all cases was a translocation between the X chromosome and an autosome, affecting the DMD gene, leading to its disruption and impaired synthesis of dystrophin protein from one allele.

Another probable contributing factor to this clinical presentation is the inactivation of the intact X chromosome, resulting in the complete absence of dystrophin expression from the second allele [12]. This mechanism maintains the required level of expression of autosomal genes during translocation between the X chromosome and the autosome [13,14]. It is known that maternal and paternal X chromosomes (Xm and Xp, respectively) have an equal probability of inactivation, which occurs early in embryogenesis, typically resulting in a 1:1 ratio of cells expressing either Xm or Xp [13].

In reciprocal X-autosome translocations, the intact X chromosome in females is usually inactivated. This helps the cell avoid monosomy for autosomal genes present on the translocated chromosome [12]. This is supported by our findings of highly skewed X-inactivation in all patients examined in this study: 96%:4%, 98%:2%, and 90%:10%.

Segregation analysis was available in all three cases, the identified balanced translocation was a de novo event, as the parents’ karyotypes were normal.

All our patients had a dystrophinopathy phenotype, characterized by predominant involvement of proximal musculature, an increase in CK levels up to 100 times the normal range, elevation of other liver enzymes up to 3–11 times, hypertrophy of the calf muscles, Gowers’ sign, and speech development delays.

In patient N1, delayed speech development was observed during the first year of life, and the ability to speak in short phrases emerged only after age 4. Examination at age 4 revealed speech limited to single words and infrequent phrases, muscle weakness, easy fatigability, and asymmetric calf hypertrophy and a positive Gowers’ sign. In patient N2, the first signs of the disease appeared during the first year of life, characterized by elevated liver enzymes (ALT, AST, and CK) and convergent strabismus. The patient also exhibited delayed speech development, weakness during physical exertion, a positive Gowers’ sign, and calf muscle asymmetry. In patient N3, the initial signs included a change in gait, tightness of the calf muscles, and a decline in academic performance. A clinical examination at age 9 revealed winged scapulae, weakness of the gluteal and thigh muscles, lumbar hyperlordosis, a waddling gait, a positive Gowers’ sign, and calf muscle hypertrophy.

As in the study by Liu C, et al., our study also described a severe form of dystrophinopathy in females with X-autosome translocations, comparable to DMD in boys [2]. However, in our case series, all girls exhibited either delayed speech development or declining academic performance. This observation is consistent with other reports.

For instance, Trippe et al. (2014) [6] described a girl with a t(X;4) translocation who presented at 4 months of age with elevated transaminases and a markedly elevated CK level (23,000 IU/L). By age 3, she exhibited proximal muscle weakness, exercise intolerance, and significantly delayed speech development, characterized by a limited active vocabulary and an inability to form two-word sentences. Zatz et al. (1981) [15] described a girl with a t(X;6) translocation. Although her early motor development was age-appropriate, by age 5 she experienced frequent falls, difficulty running, and trouble rising from a chair. By age 10, she had lost the ability to ambulate independently.

In the cases presented in the current study, all girls remained ambulatory, likely due to their young age (under 10 years at the time of examination). However, the future progression of the disease and maintenance of ambulatory status are difficult to predict, given the limited number of similar cases reported in the literature.

A study by Segarra-Casas et al. (2024) [7] described a 29-year-old female patient who had been diagnosed at approximately 10 years of age. She carried a balanced translocation, t(X;17)(p21.1;q23.2), and presented with a classic symptomatic profile, including frequent falls, difficulties with running and rising from the floor, and a significantly elevated serum CK level of up to 14,000 IU/L. Clinical examination revealed calf muscle hypertrophy and use of Gowers’ sign. She received steroid therapy from age 10 until age 25; by age 18, she was only able to walk short distances; at 21, she was diagnosed with osteoporosis; and by age 24, due to femoral and ankle fractures, she lost the ability to walk independently [7].

Pluta et al. (2023) [16] described two girls with X-autosomal translocations involving the *DMD* gene. One patient presented with cognitive impairment, proximal muscle weakness, and difficulty with physical activity. Notably, both girls exhibited delayed motor development. This contrasts with our cases, where motor development was age-appropriate. Szűcs et al. (2022) [17] described a girl with a t(X;10) translocation who, despite normal speech development, exhibited motor delays. These delays manifested as an unsteady gait and impaired balance at age 3, followed by poor coordination by age 6. According to Szűcs et al., the patient’s CK levels were markedly elevated at 9892 IU/L at age 7 and 10,694 IU/L at age 9 (reference range: 24–195 IU/L). By age 13, she exhibited a gradual deterioration in muscle strength with muscle wasting, while her CK level was 1868 IU/L [17].

Interestingly, strabismus was observed in two girls in our cohort (Patients N1 and N2), a finding not reported in other case series [6,15,16,17,18]. Collectively, these reports highlight the remarkable variability in the clinical presentation of DMD in females. This heterogeneity complicates prognosis and underscores the necessity for comprehensive diagnostic testing when standard evaluations are negative despite a strong clinical suspicion.

Speech delay and behavioral problems in symptomatic carriers of pathogenic variants in the *DMD* indicate a more severe dystrophinopathy phenotype, involving not only the muscular system but also cognitive functions. Previously conducted neuropsychological studies by Demirci et al. confirmed deficits in attention, memory, language, executive function, and visuospatial processing in some female patients with dystrophinopathy [19]. Aside from that, these children more frequently have neurodevelopmental disorders (obsessive-compulsive disorder, attention-deficit/hyperactivity disorder, autism spectrum disorders) and delayed speech development [19].

Cognitive impairments in dystrophinopathies are apparently associated with insufficient expression of specific dystrophin isoforms, such as dp427m, dp427c, and dp427p, which are full-length dystrophin transcripts with tissue-specific promoters located upstream of exon 1. These isoforms are expressed in key brain structures such as the cerebral cortex (glial cells) and the cerebellum (Purkinje cells) [20,21].

In X-autosome translocations where the breakpoint is located within the DMD gene, the formation of a full-length transcript, including the “brain” isoform dp427, is impossible from one gene copy due to disruption and spatial dislocation of the gene segments. This leads to more pronounced cognitive symptoms in cases where the *DMD* gene is affected by the translocation [7].

Although cardiac ultrasound revealed no severe functional impairments in all our patients, this finding may reflect their current young age and the activity of compensatory mechanisms. It is important to ensure regular follow-up for these patients, including routine cardiologist examinations and annual ECG monitoring. This significantly reduces risk and helps preserve long-term health, as it is known that the prevalence of cardiac problems increases with age and progresses from an asymptomatic stage (detectable on ECG) to dilated cardiomyopathy in 7% of cases [22].

Muscle MRI in one of the patients, N3 (MRI was carried out at age 8), showed a pattern characteristic of dystrophinopathy: relative preservation of the sartorius, gracilis, adductor longus, and semitendinosus muscles, also known as the “trefoil with single fruit” sign [23]. According to Zheng Y et al., this “trefoil with fruit” sign has a specificity of 99.2% for dystrophinopathies, making it a useful additional diagnostic indicator for both males and females.

In patient N1, muscle MRI revealed no significant alterations, likely due to the patient’s young age (MRI was performed at age 4). According to Zheng Y et al., the sensitivity of the “trefoil with fruit” sign for diagnosing dystrophinopathies depends on the patient’s age. Typically, patients exhibiting this sign were significantly older (mean age of 8.8 ± 4.1 years) than those without the sign. The study by Zheng Y et al. states that 63.1% of male patients aged 7–16 had the “trefoil with fruit” sign, whereas those under age 7 had it only in 18.8% of cases [23].

Currently, there is limited data regarding the evaluation of muscle MRI in females who are heterozygous carriers of a pathogenic variant in *DMD*. Consequently, no large-scale statistical studies have been conducted. However, it has been noted that muscle involvement in such cases often shows asymmetry, which differs from the classic presentation of Duchenne/Becker muscular dystrophy [24].

Given that a genetic alteration (balanced translocation) was identified in the child, it is important to assess the risk for future pregnancies. The recurrence risk of a child being born with a balanced translocation in parents with normal karyotypes is close to the general population risk and is estimated at approximately 0.2–0.5% [25].

For patients with balanced translocations disrupting the dystrophin gene, thereby fully or partially reducing protein expression, pathogenetic therapies based on the mechanism of stop codon readthrough (Translarna), as well as exon skipping therapy (Exondys), used in the treatment of Duchenne/Becker muscular dystrophy (DMD/BMD) in boys [26], are inapplicable. However, gene therapy with Delandistrogene moxeparvovec, which is currently used with restrictions in Russia for boys with a confirmed DMD diagnosis, may be appropriate for girls with severe clinical manifestations resembling the classical form of DMD.

Delandistrogene moxeparvovec is a recombinant adeno-associated viral (AAV) vector designed for the therapeutic expression of a microdystrophin gene, aiming to restore the functional dystrophin-associated glycoprotein complex in patients with DMD. It is administered as a single-dose therapy approved in Russia for patients under the age of 8 years with a confirmed pathogenic variant in the *DMD* gene, regardless of ambulatory status [27].

Given that in cases of reciprocal translocations involving the X chromosome, the intact X chromosome is most likely inactivated and no functional protein is produced from the second allele, a structurally and functionally intact copy of dystrophin is necessary to restore the function of the dystrophin-sarcoglycan complex in female patients with such translocations.

At present, symptomatic therapy is available for such patients, including cardioprotective agents and corticosteroids to alleviate muscular symptoms [28,29].

The data obtained highlight the importance of comprehensive genetic testing in women with suspected dystrophinopathy, especially in case of elevated CK levels. In the past, genomic rearrangements were identified by karyotyping, which allowed only approximate localization of breakpoints down to the sub-band level, potentially leading to diagnostic errors [30]. Underdiagnosis of DMD-like phenotypes in female patients frequently results in misdiagnosis and/or diagnostic errors in women with muscular dystrophy. Thus, in female patients with high CK levels, positive Gowers’ sign, cardiomyopathy, and no potentially causative variants detected by exome sequencing, investigation for chromosomal rearrangements should be undertaken using conventional karyotyping or WGS.

Results of MLPA and sequencing in the examined patients with X-autosome translocations were negative. Therefore, this study emphasizes the importance of a comprehensive diagnostic approach, including cytogenetic analysis (GTG-banding), as well as the incorporation of whole-genome analysis into diagnostic protocols in the absence of the most common (deletions, duplications, point variants) variants in the *DMD* gene [12].

The obtained results may contribute to earlier and more accurate diagnosis, which is particularly important for timely initiation of therapy and improved prognosis.

A limitation of this study was the unavailability of muscle biopsy material from the patients, which precluded, on the one hand, RNA-level confirmation of the absence of dystrophin expression from the rearranged gene copy, and on the other hand, immunohistochemical analysis of biopsy samples to demonstrate the absence or reduction of dystrophin directly in muscle tissue.

## 4. Patients and Methods

Three unrelated female patients were examined after consulting a clinical geneticist or neurologist at the Consultation Department of the Federal State Budgetary Scientific Institution “N.P. Bochkov Research Centre for Medical Genetics” with complaints of delayed psychomotor and speech development, predominantly proximal muscle weakness, calf muscle hypertrophy, and elevated CK levels up to 100 times the normal range (16,000–26,694 U/L). Informed consent to clinical examination and the publication of their details and images for the study was obtained from all patients or their legal representatives. This study was performed in accordance with the Declaration of Helsinki and approved by the local ethics committee of the Research Center for Medical Genetics (approval number 2021-4/1).

### 4.1. Clinical Evaluation

Clinical data, including the history of disease development, age of onset, disease onset and progression, and family history of all patients, were collected and reviewed retrospectively. Additionally, CK levels, cardiac ultrasound (US), electrocardiography (ECG), electromyography (EMG), and brain and lower limb muscle magnetic resonance imaging (MRI) data were analyzed.

### 4.2. Genetic Analysis

Whole venous blood collected in disposable EDTA-containing plastic tubes was used for the study. DNA was extracted from peripheral blood leukocytes using the Wizard^®^ Genomic DNA Purification Kit (Promega, Madison, WI, USA), following the manufacturer’s protocol [31]. Initially, to establish the cause of the disease, we used a custom panel “Limb-Girdle Muscular Dystrophies,” which included the coding sequences of the following genes: *DMD, CAPN3, EMD, SGCG, SGCA, SGCB, SGCD, TCAP, FKRP, POMT1, POMT2, ANO5, FKTN, ISPD, LMNA, CAV3, DAG1, POPDC3, FHL1, GAA, PLEC, POMGNT1, POMGNT2, GMPPB, HNRNPDL, GNE, FKBP14, DYSF, DNAJB6, BVES, and TRIM32*. The analysis was carried out on the NextSeq/MiSeq sequencer using the paired-end reading method. For library preparation, we used ultra-multiplex PCR technology followed by sequencing (AmpliSeq™ (Illumina, San Diego, CA, USA)).

Subsequently, screening for gross deletions and duplications in the DMD gene was carried out using the multiplex ligation-dependent probe amplification (MLPA) SALSA P034 and P035 DMD probemix kits (MRC-Holland, Amsterdam, The Netherlands) [32]. Reaction products were detected by fragment analysis using an ABI Prism 3500 (Applied Biosystems, Waltham, MA, USA). Data analysis was carried out using the Coffalyser.Net™ v.8 software.

In the final stage of the study, whole-genome sequencing (WGS) was performed as part of the National Genetic Initiative “100,000 + Me” [33]. Genomic DNA was extracted using a magnetic bead–based sorption method (MGIEasy Magnetic Beads Blood Genomic DNA Extraction Kit, MGI) and employed for library construction. Libraries were prepared using an enzymatic fragmentation, PCR-free protocol (MGIEasy FS PCR-Free Library Prep Set, 96 reactions (MIX), MGI). Sequencing was carried out on a DNBSEQ-T7 instrument with paired-end 150 bp reads (PE150), targeting a median coverage of 30×. Reads were passed to cutadapt 4.2 [34] for adapter removal (MGIEasy DNA Adapters) and trimming of low-quality read ends. Mapping to the reference genome GRCh38.d1.vd1 was performed with bwa 0.7.17 [35]. Duplicate marking was performed with Picard 2.27.5 [36]. Short variant calling was performed using DeepVariant 1.4 [37].

The identified variants were called according to the nomenclature standards provided at https://hgvs-nomenclature.org (accessed on 4 September 2025), version 21.1.1 [38]. Sequencing data processing was carried out using the online service “NGSData” for automated bioinformatic analysis of next-generation sequencing data [39].

To estimate the population frequencies of the identified variants, data from the 1000 Genomes Project and The Genome Aggregation Database v3.1.2 were used. The clinical relevance of the variants was evaluated using the OMIM and LOVD databases [40,41]. Variant pathogenicity interpretation was conducted according to ACMG criteria [42].

Chromosomal analysis was carried out on GTG-banded metaphases prepared from the fresh blood samples of the patients using the laboratory’s standard procedures. GTG-banded metaphase chromosomes were analyzed using AxioImager A1 microscope (Carl Zeiss, Jena, Germany) with Ikaros Karyotyping System Software, V.5.8.14 (Metasystems, Altlussheim, Germany). 11–15 metaphases were analyzed for each sample.

To determine the degree of X-chromosome inactivation, methylation-sensitive quantitative fluorescent PCR (QF-PCR) was carried out targeting X-chromosomal sequences containing tandem repeats: HUMARA and the promoter region of the RP2 gene [43], followed by fragment analysis. The X-chromosome inactivation (XCI) pattern was studied in females with dystrophinopathy by analyzing methylation of the binding site for the methylation-sensitive restriction enzyme (HpaII) in the first exon of the *AR* gene and the promoter region of the *RP2* gene.

Subsequently, the percentage ratio of cells carrying the inactivated paternal or maternal X chromosome was calculated. The XCI pattern was considered random for ratios ≤ 75:25 and skewed for ratios ≥ 75:25 [44].

## 5. Conclusions

X-linked recessive disorders can cause the development of muscular dystrophy not only in male patients. In rare cases, symptoms may also manifest in females who carry a pathogenic variant in the *DMD* gene or have a balanced translocation with a breakpoint within the *DMD* gene. In this case, the clinical presentation in female patients is caused by a reciprocal X-autosome translocation with a breakpoint in the *DMD* gene and skewed X-inactivation, likely serving as a mechanism to avoid autosomal monosomy.

In clinical practice, this implies that even in the absence of a typical family history (e.g., no affected males in the family), clinicians should not exclude X-linked predominantly recessive disorders in girls with muscle weakness. In such cases, standard genetic tests may give false-negative results if the possibility of structural chromosomal rearrangements is not considered.

This study contributes to a better understanding of the clinical and genetic heterogeneity of the disease and highlights the importance of a comprehensive approach to genetic testing in diagnostically challenging cases.

## Figures and Tables

**Figure 1 ijms-26-09389-f001:**
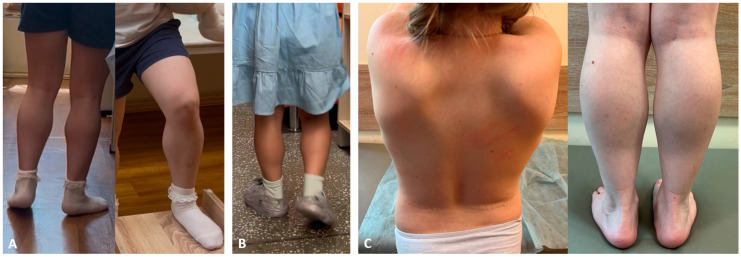
Clinical characterization of female patients with reciprocal translocations t(X;2), t(X;12), and t(X;16), with breakpoints located within the DMD gene sequence. (**A**) Patient N1 and (**B**) patient N2 demonstrated bilateral calf muscle hypertrophy, and (**С**) patient N3 exhibited bilateral scapular winging and calf muscle hypertrophy.

**Figure 2 ijms-26-09389-f002:**
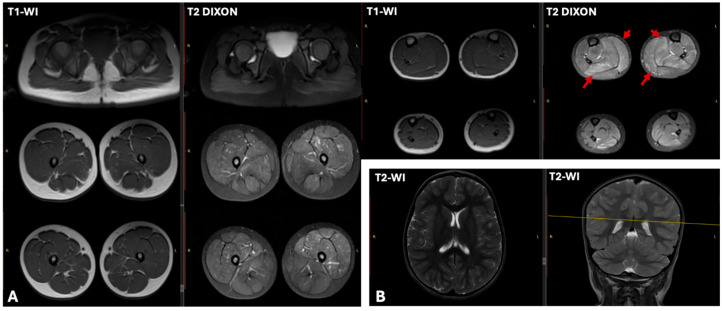
Brain and muscle MRI findings in patient N1 at age 4. (**A**) Muscle MRI of the lower limbs. T1-weighted images show no fatty infiltration. Conversely, T2 DIXON images reveal clear hyperintensity within the soleus and medial gastrocnemius muscles (arrows), indicating edema. (**B**) Brain MRI was unremarkable, with no pathological findings.

**Figure 3 ijms-26-09389-f003:**
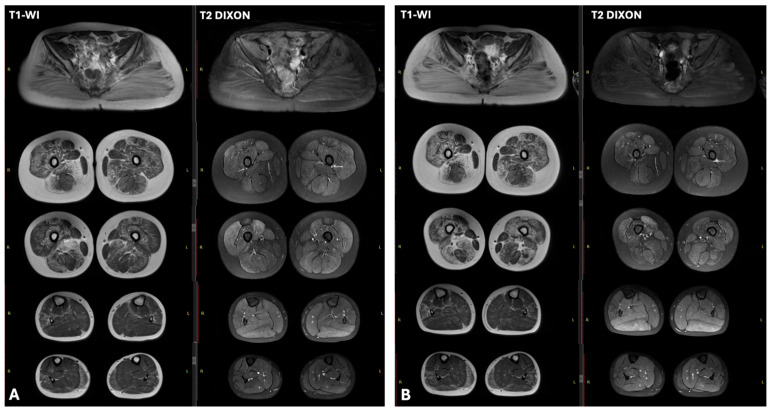
Muscle MRI of patient N3 at age 7 (**A**) and 8 (**B**). Consecutive axial T1-weighted MRI images of the thighs demonstrate a characteristic “trefoil with a single fruit” sign, with severe involvement of the adductor magnus and biceps femoris and relative sparing of the gracilis, sartorius, adductor longus, and semitendinosus. A comparison of the two time points reveals disease progression, evidenced by increased fatty infiltration within the quadriceps muscles over the one-year interval. All gluteal muscle subgroups (maximus, medius, minimus) were severely affected by fatty replacement. In the distal lower limbs, the peroneus longus and medial gastrocnemius muscles showed moderate fatty infiltration.

**Figure 4 ijms-26-09389-f004:**
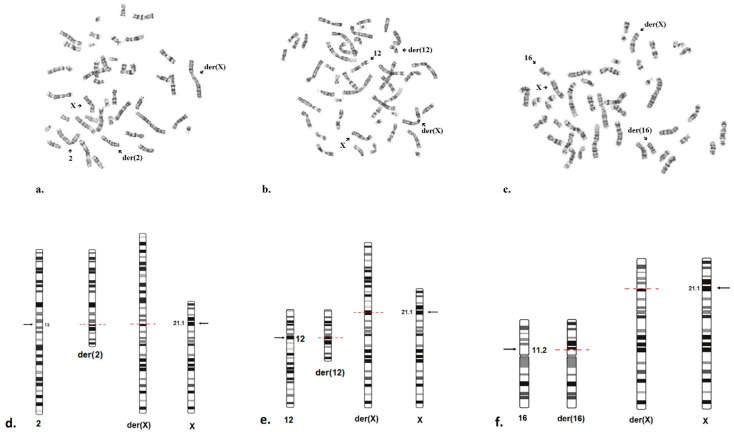
GTG-banded metaphase spreads showing the balanced translocation: (**a**) between chromosome X and chromosome 2; (**b**) between chromosome X and chromosome 12; (**c**) between chromosome X and chromosome 16. Structural rearrangement scheme: (**d**) 46,X,t(X;2)(p21;q13) in patient N1; (**e**) 46,Х,t(X;12)(p21.1;q12) in patient N2; (**f**) 46,X,t(X;16)(p21.1;p11.2) in patient N3.

**Table 1 ijms-26-09389-t001:** Clinical findings in patients.

Patient	Gender	Family History	Genotype (WGS+ Karyotype)	Segregation	Age of Examination	Age of Clinical Manifestation	First Symptoms	Delayed Walking	Proximal Weakness	Waddling Gait	Gowers’ Sign	Calf Muscle Hypertrophy	Intellectual Disability
N1	Female	no	46,X,t(X;2) (p21;q13)	de novo	4 years	1 years	Speech developmental delay, strabismus	No	Yes	Yes	Yes	Yes	Yes
N2	Female	no	46,Х,t(X;12) (p21.1;q11-q12)	de novo	6 years	6 months	Increased levels of CK, ALT and AST	No	Yes	Yes	Yes	Yes	Yes
N3	Female	no	46,X,t(X;16) (p21.1;p11.2)	de novo	9 years	5 years	Hyperlordosis, gait disturbance	No	Yes	Yes	Yes	Yes	Yes

**Table 2 ijms-26-09389-t002:** Laboratory and instrumental examination in a cohort of female patients with translocations.

Patient	CK (U/L)	ALT and AST (U/L)	ECG	Cardiac US	Lower Limb Muscle MRI	EMG
N1	26,694	x11	Moderate sinus arrhythmia, incomplete RBBB. Deep Q wave in leads III and aVF. Slight impairment of repolarization in the myocardium, manifested as T wave flattening in lead III	LVEF—61.4% (normal >60%), LVEDD—37.4 mm (normal <37 mm).	T1-WI showed no fatty infiltration. T2 DIXON images revealed clear hyperintensity within the soleus and medial gastrocnemius muscles	myogenic pattern
N2	23,000	x6	Incomplete RBBB, moderate hypoplasia of the pulmonary artery trunk	Ebstein’s anomaly	No data	myogenic pattern
N3	16,000	x3–5	Abnormal ventricular repolarization	No changes	Characteristic “trefoil with a single fruit” sign, with severe involvement of the adductor magnus and biceps femoris and relative sparing of the gracilis, sartorius, adductor longus, and semitendinosus.	myogenic pattern

## Data Availability

The data presented in this study are available upon request from the corresponding author.

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
