# Peer review of "Balanced Translocations Involving the DMD Gene as a Cause of Muscular Dystrophy in Female Children: A Description of Three Cases"

_ijms, 2025, doi:10.3390/ijms26199389_

Round 1

Reviewer 1 Report

Comments and Suggestions for Authors

Although cases of Duchenne muscular dystrophy (DMD) in females are rare, several have been described in the literature. While this article presents three cases, the novelty of the report should be highlighted to emphasize its originality: simply documenting rare cases is not inherently novel.

For example, the authors should delve deeper into the clinical aspects of each case. For example, in one patient, delayed speech development was observed during the first year of life, and the ability to speak in short phrases emerged only after age 4. Divergent strabismus on the right side was observed from age 1, along with behavioral peculiarities such as limited social interaction; although she understood spoken language, she did not follow commands. Examination at age 4 revealed speech limited to single words and infrequent phrases, muscle weakness, easy fatigability, and asymmetric calf hypertrophy. The authors should clarify whether these features (e.g., neurodevelopmental delays, strabismus, and asymmetric calf hypertrophy) are common or uncommon in female DMD cases, based on the relevant literature for their comparison. Similar analyses should be extended to the other two patients to strengthen the discussion.

Author Response

Comments 1: The authors should clarify whether these features (e.g., neurodevelopmental delays, strabismus, and asymmetric calf hypertrophy) are common or uncommon in female DMD cases, based on the relevant literature for their comparison. Similar analyses should be extended to the other two patients to strengthen the discussion.
Response 1: Thank you for your review and valuable feedback on our manuscript. We agree that comparing the clinical presentations would strengthen the paper. As suggested, we have now expanded the discussion to include a comparative analysis with relevant clinical cases from the literature.
The significant clinical symptoms observed in our patients are now summarized in lines 259-270. Furthermore, we have added a comparative analysis with cases from the literature in lines 273-282 and 296-311.

Reviewer 2 Report

Comments and Suggestions for Authors

In the reviewed manuscript the authors describe a series consisting of three female patients with suspected muscular dystrophy. Following the negative results of exome sequencing and MLPA analysis - the current standard of care, based on clinical suspicion for genetic etiology, genome sequencing was also performed. Genome sequencing revealed possible translocations between X chromosome and autosomes with breakpoints within DMD gene in all cases, which was confirmed using classical karyotyping. The authors further proved highly skewed X-inactivation in all three cases. The authors also proved de-novo ocurence of the translocation in two cases, while the parents were not available for testing in one case. The authors present clear and thorough clinical information on all three patients focusing on the presentation of the disease in females, which is one of the main strengths of the article. Additionally, the authors theorize on the reasons for skewed X inactivation and highlight the importance of using comprehensive methods such as genome sequencing to detect rare variant types such as copy neutral structural variants.

The articles does a good job of covering the previously published knowledge on translocations being a rare pathogenic variant type in one of the most common hereditary muscular dystrophies, DMD, and contributes valuable knowledge to the clinical and laboratory genetics communities. In the current landscape of exome sequencing based rare disease diagnostic it is highly relevant to spread awareness about the limitations of the method, which is what this article also does well. Finally, describing a case series instead of a single case will also contribute to the article being more impactful.

In terms of room for improvement, the discussion appears to stand out as the weakest due to the authors decision to focus on the laboratory aspects of these three cases first and the clinical observations second, while in the previous parts of the article the focus was on the clinics. I would suggest reformatting this part of the article to fit with the results section in terms of following the same conceptual flow. Additional minor comments are below: 

In Table 1 there is conflicting information between table - all three variants are marked as de-novo, and text where it is explained that one family (patient N2) was not available for segregation analysis. Please correct the data in the table or text.

In lines 181/182, the authors should not classify the variant in TRAPPC11 as a VUS, instead it should be reported as carriership of a single pathogenic variant in a gene associated with recessive disorders.

In lines 214/230, the authors use valuable space at the beginning of the discussion to revisit a VUS and carriership of a pathogenic variant in two patients. This could be summed up in one sentence similar to line 213, e.g. "no other clinically relevant variants were found in the three patients". This space could perhaps be better used to comment on the importance of considering copy neutral changes in cases of DMD with negative exome sequencing and MLPA analysis as well as the importance of using genome sequencing in cases with a high likelyhood of genetic etiology for the disease.

In lines 433/435 the authors should rewrite the second sentence of conclusion for clarity - a balanced translocation with breakpoints in DMD would be considered a pathogenic variant. 

Author Response

Comments 1: In Table 1 there is conflicting information between table - all three variants are marked as de-novo, and text where it is explained that one family (patient N2) was not available for segregation analysis. Please correct the data in the table or text.

Response 1: Thank you for this important comment. You are correct. The karyotype for the third family was indeed performed, but we received this information after the initial analysis of the other cases. We have now included it in the revised manuscript. «For all three patients, karyotype analysis revealed that the translocations were balanced. It was determined that the karyotypes of the parents of patients N1, N2 and N3 were normal».  Lines 198-200. The correction has also been made in the abstract (Lines 31-32). Karyotyping confirmed reciprocal translocations in all patients, with de novo status established in all three cases.

Comments 2: In lines 181/182, the authors should not classify the variant in TRAPPC11 as a VUS, instead it should be reported as carriership of a single pathogenic variant in a gene associated with recessive disorders.

Response 2:   We thank the reviewer for this valuable comment regarding the variant classification. We agree with this insight and have reclassified the variant as representing a carrier state in the revised manuscript. «This variant was inherited from the mother of proband 3. No second pathogenic or likely pathogenic variant was found, and the variant was classified as carriership of a single pathogenic variant in a gene associated with recessive disorders.»

Comments 3: In lines 214/230, the authors use valuable space at the beginning of the discussion to revisit a VUS and carriership of a pathogenic variant in two patients. This could be summed up in one sentence similar to line 213, e.g. "no other clinically relevant variants were found in the three patients". This space could perhaps be better used to comment on the importance of considering copy neutral changes in cases of DMD with negative exome sequencing and MLPA analysis as well as the importance of using genome sequencing in cases with a high likelyhood of genetic etiology for the disease.

Response 3: Thank you for this important comment regarding the interpretation of heterozygous variants identified by WGS in girls with a clinical picture of DMD. We have considered this point carefully. We believe that ruling out other variants associated with an autosomal recessive inheritance pattern is important when performing and interpreting WGS in female. All other causes of the disease must be excluded. We would like to keep this discussion.

Comments 4: In lines 433/435 the authors should rewrite the second sentence of conclusion for clarity - a balanced translocation with breakpoints in DMD would be considered a pathogenic variant.

Response 4: Thank you for this suggestion. We have addressed it in the revised manuscript (lines 472-474). In rare cases, symptoms may also manifest in females who carry a pathogenic variant in the DMD gene or a balanced translocation with breakpoints in DMD would be considered a pathogenic variant.